

*Manuscript for Biogeosciences*
**Title: Evaluation of modeled global carbon dynamics: analysis based on global**
**carbon flux and above-ground biomass data**
**Running Title:** Evaluation of modeled global carbon dynamics
Bao-Lin Xue[1], Qinghua Guo[1,2,*], Tianyu Hu[1], Yongcai Wang[1], Shengli Tao[1], Yanjun
Su[2], Jin Liu[1] and Xiaoqian Zhao[1]
[1]State Key Laboratory of Vegetation and Environmental Change, Institute of Botany,
Chinese Academy of Sciences, No. 20 Nanxincun, Xiangshan, Beijing 100093, China
[2]School of Engineering, Sierra Nevada Research Institute, University of California at
Merced, CA 95343, USA
**Author Agreement:**
All authors agree on the authorship of the manuscript and approve the submission.
**\*Corresponding author:**
Qinghua Guo, Dr., Prof.
State Key Laboratory of Vegetation and Environmental Change, Institute of Botany,
Chinese Academy of Sciences, No. 20 Nanxincun, Xiangshan, Beijing 100093, China
Tel.: +86-10-6283-6473
Email: guo.qinghua@gmail.com





**Abstract**
Dynamic global vegetation models are useful tools for the simulation of carbon
dynamics on regional and global scales. However, even the most validated models are
usually hampered by the poor availability of global biomass data in the model
validation, especially on regional/global scales. Here, taking the integrated biosphere
simulator model (IBIS) as an example, we evaluated the modeled carbon dynamics,
including gross primary production (GPP) and potential above-ground biomass
(AGB), on the global scale. The IBIS model was constrained by both in situ GPP and
plot-level AGB data collected from the literature. Independent validation showed that
IBIS could reproduce GPP and evapotranspiration with acceptable accuracy at site
and global levels. On the global scale, the IBIS-simulated total AGB was similar to
those obtained in other studies. However, discrepancies were observed between the
model-derived and observed spatial patterns of AGB for Amazonian forests. The
differences among the AGB spatial patterns were mainly caused by the
single-parameter set of the model used. This study showed that different
meteorological inputs can also introduce substantial differences in AGB on the global
scale. Further analysis showed that this difference is small compared with
parameter-induced differences. The conclusions of our research highlight the
necessity of considering the heterogeneity of key model physiological parameters in
modeling global AGB. The research also shows that to simulate large-scale carbon
dynamics, both carbon flux and AGB data are necessary to constrain the model. The



main conclusions of our research will help to improve model simulations of global
carbon cycles.

**Keywords**: dynamic global vegetation model, integrated biosphere model, gross
primary production, above-ground biomass, global carbon cycle



## 1 Introduction


The global terrestrial ecosystem is an important carbon sink that can mitigate the
ongoing increases in atmospheric $CO_2$ concentration (Dixon et al., 1994; Luyssaert et
al., 2007; Pan et al., 2011). For example, global forests, which cover around 30% of
the land surface, account for ~75% of terrestrial gross primary production (GPP) and
~80% of global plant biomass (Kindermann et al., 2008; Beer et al., 2010). The large
carbon stock in the terrestrial ecosystem indicates the need for a reliable description
of its current distribution and prediction of future variations (Keith et al., 2009;
Galbraith et al., 2010; Pan et al., 2011; Xue et al., 2011). However, it is still a
challenge to accurately estimate the distribution of carbon stocks on the global scale,
mainly because of the unknown mechanisms and/or relative contributions of various
factors such as climate change, $CO_2$ fertilization, and land use change on carbon
dynamics (McGuire et al., 2001; Mu et al., 2008).
Various methods have been developed for mapping the global distribution of
biomass, and each has its pros and cons. On the regional scale, the field inventory
method provides the most reliable information on biomass, but it is labor intensive
and costly when applied over a large area (e.g., Malhi et al., 2002). On the global
scale, remote-sensing methods have advantages over field inventory methods for
applications to large areas and in areas that are difficult to access (Lefsky et al., 2005;
Thurner et al., 2014; Tao et al., 2014). For example, the light detection and ranging
method has recently been used in the Amazon region, with acceptable accuracy



(Asner et al., 2010; Saatchi et al., 2011). As an alternative, the dynamic global
vegetation model (DGVM) is a useful tool for mapping global biomass and is the only
method that predicts future variations. In the past, many researchers have explored
how climate change or land use change would alter the global biomass, and this has
improved our confidence in the projection of terrestrial responses to climate change.
In many cases, the DGVM-modeled potential vegetation biomass is used as a baseline
for exploring the corresponding response to the projected climate. Before using the
DGVM to project future biomass changes, an evaluation of how the DGVM can
reproduce potential (natural) present-day biomass is necessary (Mu et al., 2008; Seiler
et al., 2014). However, this is rarely done, mainly because of the lack of available
global-scale biomass data. For instance, in many cases, the default values for various
physiological parameters are used, and may differ greatly for different DGVMs. The
lack of evaluation of modeled biomass on the global scale may result in large
differences among global carbon stocks obtained using different models (Cramer et al.,
2001; Sitch et al., 2008), resulting in bias in our conclusions regarding vegetation
responses in projected climate scenarios (Huntingford et al., 2008; 2013).
Uncertainty in the modeled biomass may originate in various ways: model
structure, model parameters, and meteorological inputs. The results for potential
natural vegetation obtained from bioclimatic limits and forest dynamics using the
DGVM may give an unrealistic representation of competition among plant functional
types (PFTs) (Purves and Pacala, 2008; Seiler et al., 2014). A biased PFT in the
DGVM partly contributes to the uncertainty in carbon dynamics, including GPP and



biomass. Moreover, DGVMs usually use a single set of parameters to represent
different biomes and rarely consider spatial heterogeneity (Xiao et al., 2011, 2014). In
reality, different physiological parameters vary greatly, depending on the soil type,
climate, and vegetation (Castanho et al., 2013). The ways in which this will bias the
spatial pattern of carbon flux, and thus biomass accumulation, have not been
sufficiently discussed on the global scale, partly because of the unavailability of
biomass data for large areas (Delbart et al., 2010; Wolf et al., 2011). Recent research
has shown that it is necessary to use both carbon flux data and biometric data for
DGVM calibration (Kondo et al., 2013; Seiler et al., 2014). Furthermore, uncertainties
in DGVM-derived carbon flux and biomass may also arise from the input data itself,
such as meteorological forcing data (Barman et al., 2014a, b). Different input data can
result in differences among the results obtained using different models when modeling
large-scale carbon flux (Zhao et al., 2005; Jung et al., 2007). It is therefore necessary
to quantify the uncertainty from meteorological inputs in modeled biomass, to
improve the modeling results.

The objective of this study is to evaluate model-derived carbon flux and biomass

on the global scale using collected carbon flux (GPP) and biomass datasets at the plot
level. To do this, we used the integrated biosphere simulator (IBIS; Foley et al., 1995;
Kucharik et al., 2000) as an example, and used both carbon flux and collected
above-ground biomass (AGB) data (2101 plots) to constrain the model. We adopted
the most important parameters from meta-analysis, calibration, or from the literature.
We also investigated how different meteorological input data changed the modeling





results. Overall, the intention of the current study was to explore the following
questions. 1) How accurately can IBIS simulate GPP and AGB, and where does bias
originate? 2) Can a single set of calibrated parameters accurately map the patterns of
GPP and AGB? 3) What should modelers do to improve the modeling results?
**2 Material and methods**
**2.1 IBIS model**

The IBIS model considers the composition and structure of vegetation responses

to environmental changes, within an integrated framework, to simulate land surface
hydrothermal processes, biogeochemical cycles, and terrestrial vegetation dynamics.
The model simulates the land surface processes for energy, water, and momentum
exchange between soil, vegetation, and the atmosphere, using a land surface transfer
scheme (LSX) (Thompson and Pollard, 1995a, b). In detail, two canopy layers, three
snow layers, and six soil layers are considered in each grid unit. Evapotranspiration
(ET) consists of three components, i.e., canopy transpiration, interception, and
evaporation from the ground surface. Vegetation transpiration is calculated using a
semi-mechanistic model of stomatal conductance (Ball et al., 1986), which is coupled
with canopy carbon assimilation and water exchange between a leaf and the boundary
layer to give
$$g_{s,h_2o} = \frac{mA_n}{C_s} h_s + b \quad (1)$$
where $A_n$ is the net photosynthesis rate at leaf level ($\mu mol\ CO_2\ m^{-2}\ s^{-1}$), $g_{s,h2o}$ is the





leaf-level stomatal conductance of water vapor (μmol $H_2O$ m$^{-2}$ s$^{-1}$), $C_s$ is the $CO_2$
concentration (μmol μmol$^{-1}$) at the leaf surfaces, $h_s$ is the relative humidity at the leaf
surface (%), and $m$ and $b$ are empirical parameters.

IBIS represents natural vegetation using PFTs, based on the biomass and leaf area

index. Overall, 12 PFTs are defined in IBIS, related to bioclimatic limits, and
physiological, morphological, phenological, and life-history criteria governing
competition for light and water (Alton, 2011). Different physiological parameters are
set for each PFT to quantify factors such as the phenological performance or carbon
assimilation and water consumption characteristics (Kucharik et al., 2000). As a result,
the GPP, and thus the net primary production (NPP) and vegetation transpiration, are
calculated separately for upper (trees) and lower (shrublands and grass) canopies as
$$NPP = (1-\eta)\int (A_g - R_{leaf} - R_{stem} - R_{root})dt \quad (2)$$
where $A_g$ is the gross canopy production, $\eta$ is the fraction of carbon lost by growth
respiration (fixed value of 0.3), and $R_{leaf}$, $R_{stem}$, and $R_{root}$ are leaf, stem, and root
respiration, respectively.

The model allows for the coexistence of different PFTs in a single grid cell.

However, a dynamic vegetation mechanism is used to simulate annual changes in
vegetation structure through PFT competition for light, water, and other nutrient
resource pools (Kucharik et al., 2006). The competition among PFTs is driven by
differences among carbon balances resulting from phenology, leaf form, and
photosynthetic pathways (Foley et al., 1996; Kucharik et al., 2000). On the annual
scale, the NPP is allocated among three carbon pools, i.e., leaves, stems (for trees),



and roots. The instantaneous change in the biomass pool $j$ of PFT $i$ is represented as
$$\frac{\partial C_{i,j}}{\partial t} = a_{i,j} NPP_i - \frac{C_{i,j}}{\tau_{i,j}} \quad (3)$$

where $a_{i,j}$ is the fraction of annual NPP allocated to the biomass pool and $\tau_{i,j}$ is the
carbon residence time of that biomass pool. Note that $a_{i,j}$ is a fixed value in IBIS, but
in some other DGVMs (e.g., the Lund–Potsdam–Jena dynamic global vegetation
model, Sitch et al., 2003) the NPP is allocated using allometric equations.
A relatively simple phenology module based on accumulated growing degree
days (Botta et al., 2000) is used in the original IBIS. A modified version of the
phenology scheme, based on that reported by Jolly et al. (2005), was developed in this
study. In detail, the prognostic phenology model is based on the growing season index
(GSI), which is decided by three main environmental factors, i.e., temperature,
photoperiod, and humidity (Equation 4). The photoperiod is calculated according to
the latitude of the model grid and empirical algorithms. We also adopted a 21-day
running mean GSI calculated from daily mean meteorological variables, following
Jolly et al. (2005).
$$GSI = f(\overline{T_m}) \times f(\overline{R_g}) \times f(\overline{VPD}) \quad (4)$$

1721. where $T_m$, $R_g$, and VPD are multi-day running mean averages of air temperature (℃),

solar radiation (W m$^{-2}$) and vapor pressure deficit (Pa); $f(\overline{Tm})$, $f(\overline{Rg})$, and $f(\overline{VPD})$
vary linearly between the constraining limits 0 and 1, and thus regulate vegetation
activity; these functions are defined in Equations (2–4) in Stöckli et al. (2008).



**2.2 Model input data**

In the present study, IBIS was executed globally at a $0.5° \times 0.5°$ latitude–longitude grid resolution. The initial vegetation type was obtained from moderate-resolution imaging spectoradiometer (MODIS) MOD12Q1 product (Friedl et al., 2010), and resampled to $0.5°$. Soil texture data were obtained from the Center for Sustainability and the Global Environment (http://www.sage.wisc.edu/download/IBIS/ibis.html), and was reformatted from the Global Soil Data Products CD-ROM issued by the International Geosphere–Biosphere Programme Data and Information Services. The topographical data were obtained from the Shuttle Radar Topographic Mission (http://srtm.usgs.gov/), with a resolution of 1000 m. We resampled the resolution to $0.5°$ (~ 50 km) as a model grid.

The climate data, including monthly mean air temperature, precipitation, relative humidity, cloudiness, diurnal temperature range, wind speed, and the number of wet days, were obtained from the Climate Research Unit (CRU) climate dataset for 1901 through 2010 (CRUTS3.10, Harris et al. 2013, hereafter CRU). We examined the modeled biomass uncertainty induced by different meteorological datasets using forcing data from Princeton University (http://hydrology.princeton.edu/data.pgf.php, hereafter Princeton) to drive the model. Princeton does not include wind speed, therefore we use the wind speed data from the Global Land Data Assimilation System covering the period 1948–2010 (http://disc.sci.gsfc.nasa.gov/services/grads-gds/gldas). The Princeton was developed at a global spatial scale of $0.5°$, with a daily timescale.



In both cases, we spun-up the model for 400 years and then conducted transient
simulations starting from 1948, and 1901 climate data were used for the years before

1901.

**2.3 Model validation data**

To calibrate and validate the IBIS model, we collected site-level GPP and ET data

from Fluxnet (http://fluxnet.ornl.gov/). The validation sites and data were carefully
selected; we only collected sites with at least 3 years' data, because there may be
greater uncertainty for sites that cover only 1 or 2 years. Thirty-nine sites were
selected, covering tropical, temperate, and boreal forests, and grasslands or croplands
(Fig. S1, Table S1). Note that IBIS does not simulate croplands explicitly; therefore
croplands were compared with the simulation results for the understory. The
calibration and validation were conducted on both monthly and annul scales.

To constrain the model with both flux and biometric data, we also collected

plot-level AGB data from the literature. Overall, 2101 site-year biomass data were
obtained on the global scale (Fig. S1, Table S1). The resolution of plot-level data is
usually 0.01 °, therefore we used the average value as a proxy for a model grid. We
also evaluated the modeled AGB on the regional scale. In detail, we first generated a
regional AGB map for tropical Amazonian forests using collected plot data (~ 400
plots) by the random forest method (Breiman, 2001); the data were then resampled at
0.5 ° for comparison with the modeling results. Note that the model calculates the
carbon density (Mg C ha$^{-1}$) instead of the AGB, therefore we calculated the AGB (Mg



ha$^{-1}$) by multiplying by a factor of 2.0 (IPCC, 2003).

## 3 Results

To minimize the number of parameters for calibration, we used most of the
default values, as in Foley et al. (1996) and Kucharik et al. (2000); we calibrated the
parameters most sensitive to the GPP and ET (Table 1). We mainly calibrated the
photosynthesis capacity at 15 ℃ (vmax_pft) for different PFTs, as in Castanho et al.
(2013). The flux data were mainly for boreal and temperate forests and grassland
(including crops), because of the gaps in data for tropical forest. We therefore used the
literature value for tropical forest (Zhu et al., 2011). Furthermore, we validated the
GPP and ET on the annual scale globally, by comparison with other released datasets.

### 3.1 Monthly-scale calibrations

The model performs well for most sites after calibration (Table 2). The Taylor
diagram shows a high correlation between the modeled and observed values for both
GPP and ET (Fig. 1). Most sites have correlation coefficients above 0.6 for GPP and
ET on the monthly scale. The model performance for ET is better than that for GPP,
with large correlation coefficients and larger determination coefficients, averaged as
0.60 and 0.74, respectively, for 39 sites (Fig. 1 and Table 2). This shows that the
model can simulate the energy balance well, according to the LSX land surface
subsection. The model simulates upper canopy (forests) better than lower canopy
(shrubs and grasses), with large correlation coefficients and small deviations from 1



for the GPP slope (Fig. 1 and Table 2).

**3.2 Annual-scale validations**

We compared our simulated GPP and ET results with annual-scale in situ
observations (Fig. 2). There are strong relationships between the model simulation
and in situ values for both GPP and ET ($R^2 = 0.57$, $p < 0.001$ and $R^2 = 0.64$ $p < 0.001$
for GPP and ET, respectively). In both cases, the simulations slightly overestimate
small values with large intercepts and slightly underestimate large values compared
with the in situ observations. This overestimation of low values is clearly seen in
independent validation by collected GPP from the literature (Fig. 2c). When the GPPs
were below 500 gC m$^{-2}$ year$^{-1}$, the simulated GPPs were around twice the observed
values. This systematic error may be caused by differences between the flux tower
fetch and the model grid resolution (Kim et al., 2006). Another reason may be that the
flux tower generally focuses on high-production ecosystems (Turner et al., 2006).

**3.3 Global annual-scale validations**

We further validated the simulated annual GPP and ET results with those from
Jung et al. (2011), on the global scale (Fig. 3). The GPP and ET were scaled up from
flux tower values using the machine-learning technique reported in Jung et al. (2011),
at the same resolution as our model grid (0.5 ° × 0.5 °). The modeled global average
GPP is 1112 gC m$^{-2}$ year$^{-1}$ for 2000–2010; this is larger than the value reported by
Jung et al. (2011) (933 gC m$^{-2}$ year$^{-1}$). The corresponding total global GPP during this



period is 142 PgC year$^{-1}$ for the model simulation. The modeled GPPs for Amazonian
and African tropical areas are usually above 2800 gC m$^{-2}$ year$^{-1}$, whereas the value
for tropical forests in southeastern Asia are usually above 3200 gC m$^{-2}$ year$^{-1}$. Our
model simulation values are ~200 gC m$^{-2}$ year$^{-1}$ larger than those reported by Jung et
al. (2011) for most areas, especially for areas with small GPPs (Fig. 3b). This
difference is even larger in southern China and the southern US. In contrast, the GPP
is less than that reported by Jung et al. (2011) for southern Amazonian areas.

Similar patterns to those for GPP are found for ET in the model simulations. The

global average ET is 449 mm year$^{-1}$, compared with the value of 546 mm year$^{-1}$
reported by Jung et al. (2011). In most areas, the model simulation results are around
100 mm year$^{-1}$ smaller than those from Jung et al. (2011), especially for low ET areas
(Fig. 4). However, the modeled ET is around 200 mm year$^{-1}$ larger than that obtained
by Jung et al. (2011) for Amazonian and southeastern Asian tropical areas (Fig. 4b).
**3.4 Plot-level biomass calibrations**

Fig. 5 shows a comparison of the modeled biomass with plot-level observations

after calibration. Fig. 5a shows all the site-year data for each plot, and Fig. 5b shows
the grid-averaged comparisons. The simulations show strong correlations in both
cases. The regression is better for the grid-level case. The improved regression
relationship in the grid-level comparison is caused by the scale difference between the
site location (0.01 °) and the model grid (0.5 °). In both cases, the model overestimated
low values but underestimated large ones. As stated in section 2.3, the plot accuracy is



usually 0.01°, therefore the modeled values seem "saturated" in some cases, as
observations vary if they are within the same grid. Fig. 5c shows an independent
validation of the modeled biomass by plot-level observations. The plots are mainly
from measured AGB from natural forests in China. The regression relationship is
significant, but also has large scattering in the calibration. Overall, the model seems to
underestimate large values, but overestimate small values (below 50 Mg ha$^{-1}$).
**3.5 Global and regional AGBs**
Fig. 6a shows the spatial pattern of the model-derived above-ground global
biomass (upper and under layers). The global average biomass is 81.73 Mg ha$^{-1}$, with
the largest values in tropical areas and the lowest in boreal areas. The global map of
AGB shows large heterogeneity, which is similar to the case for global GPP patterns.
The zonal AGB within each 0.5° latitude interval shows a large fluctuation (Fig. 6b).
The AGB is relatively small below −30°S and then starts to increase sharply to a
maximum of 278.44 Mg ha$^{-1}$ at around −1.25°S (AGB = 10.814 × latitude + 291.03,
$R^2 = 0.95$, $p < 0.001$). The AGB then decreases sharply until 13.75°N (AGB = −8.04
× latitude + 313.3, $R^2 = 0.95$, $p < 0.001$). The AGB is relatively constant between 15°
N and 50°N and then increases. The AGB reaches another maximum, 112.17 Mg ha$^{-1}$,
at around 56.25°N, and decreases continuously to close to 0 at around 75°N.
Fig. 7 shows a comparison of the regional AGBs for Amazonian tropical forests.
The observed regional AGBs are derived from 399 plot-level data using a random
forest method. The calculated average AGB is 280.27 Mg ha$^{-1}$, and shows a



decreasing gradient from east to west. The model calculates the average AGB in this
area as 285.95 Mg ha$^{-1}$, which is comparable to the observed value. However, our
modeled AGB does not show a decreasing gradient from east to west, but shows a
decreasing gradient from north to south gradient as that for GPP (Fig. 6). The model
therefore underestimates the large AGB in the east and overestimates the AGB in the
west (Fig. 7b). Most grids in the Amazonian region are within a ±30% relative error
[(Model − Observation)/Observation × 100%) (Fig. 7b). This results in a small
absolute error of 4.42 Mg ha$^{-1}$ over the whole area.

**3.6 Global AGB driven by CRU metrological data**

Fig. 8 shows the spatial pattern of the difference between AGBs driven by
Princeton and CRU. Most areas of the globe show AGB differences within 20 Mg
ha$^{-1}$, according to the two meteorological datasets. The average global difference is
12.83 Mg ha$^{-1}$, with large heterogeneities in different areas. Large differences are
observed in savanna regions (MODIS UMD classification scheme) in South America
and central Africa, and shrublands in northeastern Siberia (Fig. 8a). In these areas, the
AGB driven by daily meteorological data (Princeton) is significantly larger than those
derived from CRU data. In contrast, in most tropical areas, the AGB derived from
Princeton datasets is smaller than those derived from CRU datasets. Most of the grids
show a relative error within ±20% with largest frequency occurs for relative error of
10 % (Fig. 8b).





**4 Discussion**


We used a single set of model parameters to estimate the global carbon stock in
terms of AGB. The IBIS model does not calculate the global AGB directly, but
calculates the carbon density. We therefore compared our model-derived carbon
density with those from other studies. Comparisons of carbon densities have the
advantage over AGB comparisons that they eliminate the uncertainties induced by
global vegetation areas used in different studies. Our model-derived carbon density is
smaller than that reported by Pan et al. (2011) on the global scale (82.96 compared
with 94.2 Mg C ha$^{-1}$), and this results in a smaller global carbon stock (Table 3). Pan
et al. (2011) calculated the carbon density, using the forest inventory method, for the
period 1990–2007; their estimated value of 94.2 Mg C ha$^{-1}$ includes both above- and
below-ground biomass. Previous research showed that ~80% of the total biomass is in
AGB and ~20% is in below-ground biomass for forest ecosystems on the global scale
(Cairns et al., 1997). This indicates that the global above-ground carbon density is
~75 Mg C ha$^{-1}$ for Pan et al. (2011). This value is comparable to our modeling result.
The difference between the global carbon stocks in AGB may arise from the different
forest areas used by Pan et al. (2011) and in our study (MODIS derived). The forest
areas were 3851.3 $\times 10^6$ and 3332.35 $\times 10^6$ ha in Pan et al.'s study and our study,
respectively. Further comparison of the regional-scale carbon density with those from
three other studies show that values in our study and those reported by Pan et al.
(2011) are larger. The carbon densities reported by Goodale et al. (2002) and Liski et
al. (2003) are around 30% smaller than those reported by Thurner et al. (2014) and in



our study for European forests. In contrast, for North American forests, the carbon
densities reported by Pan et al. (2011) and in our study are similar, and larger than
those in the other three studies. These comparisons with other studies show that the
IBIS-model-derived carbon density gives reasonable results on the global scale and
can therefore be used as an independent tool for validating AGB estimations by other
methods.

A regional-scale comparison of the observed and modeled AGBs for Amazonian

tropical forests shows that the spatial patterns in the modeling results are biased (Fig.
7). The relative error between the modeled and observed GPPs in this region is
usually below 10% (Figs. 2 and 3). However, the relative error in the AGB for most
grids is within ±30% (Fig. 7). This indicates that the uncertainty in the modeled AGB
may be mainly caused by woody carbon residence ($\tau_w$, Table 1) instead of carbon
assimilation. Though our point-level calibration shows a significant relationship
between modeled and plot level data, the calibration points are subject to scatter.
Independent validation shows that the model tends to underestimate the AGB when
the AGB is large (Fig. 5c). Similar determination coefficients ($R^2$) were reported by
Seiler et al. (2013) for a regional-scale model calibration in Bolivia. The relatively
small $R^2$ may explain the region-scale difference for Amazonian forests. The single
value of $\tau_w$ in the model cannot reproduce the spatial variance of AGB on a large scale.
Similar research by Castanho et al. (2013) showed that the woody biomass residence
time is the most important parameter in determining the spatial variance in modeled
AGB in this area. Further investigation using a spatial pattern of $\tau_w$ in IBIS greatly





improved the modeled AGB, with $R^2$ changing from 0.33 to 0.88 (Castanho et al.,
2013). These and the presented results indicate that to improve the model simulation
accuracy, modelers should consider the spatial heterogeneity of the most important
parameters in the model used, especially for large-scale simulations (e.g., Zhou et al.,

2009).

Climate-data-driven uncertainties in modeling carbon and energy cycles have
previously been analyzed (Zhao et al., 2005; Barman et al., 2014a, b). A systematic
analysis based on various global vegetation models and meteorological data showed
that substantial changes in the modeled GPP were observed for different
meteorological inputs in regional simulations in Europe (Jung et al., 2007). The
interannual variations in the GPP were mainly caused by different meteorological
drivers. A similar analysis by Barman et al. (2014b) showed that the differences in
site-level GPPs caused by different meteorological drivers were ~20% of the annual
GPP. This was mainly caused by biases in short-wave radiation and humidity for
various meteorological drivers tested in the study. Our study results show that
climate-data-driven uncertainties in carbon assimilation (GPP) can be transferred to
the AGB carbon stock (Fig. 8). The relative differences caused by different climate
drivers are generally within ±20% (Fig. 8b). These differences are smaller than the
relative errors induced by the invariant parameters over the Amazonian forest. This
indicates that to improve the model simulation accuracy, modelers should pay
attention to both model parameter calibration and metrological drivers, with a focus
on the former.



Data availability is one of the main reasons that few global model simulations use
plot-level data to constrain the model (Seiler et al., 2014). We collected plot-level
AGB data from the literature, and used them to calibrate and validate IBIS on the
global scale. The plot resolution was generally 0.01–0.1 ° (~1–10 km). In the
validation, we used measured single-point values as a proxy for a model grid average
(~2500 km$^2$), which may have caused a bias relative to the modeled values. Note that
even over a small area, AGB may vary greatly because of local soil type, land use
variability, and local water availability (Baker et al., 2004). Therefore, the difference
between the spatial scales of the plot level and our model simulation grid may partly
explain the small $R^2$ in Fig. 5. Further investigations of model simulations at different
spatial resolution (especially at high resolution) are therefore necessary to facilitate
model calibration by higher spatial resolution AGB datasets. Furthermore, the plot
points used for validation and calibration are from natural forests, with little human
disturbance, therefore our modeling results represent the potential value under current
climate conditions (e.g., Mu et al., 2008; Seiler et al., 2014). The AGBs in Table 2 are
present-day AGBs, which may be influenced by human activities. A direct comparison
of model simulation and these data is therefore to some extent inappropriate. However,
this comparison is useful, because based on exploration of the difference between the
two, the model could be used to quantify the impact of human activities (such as land
use change, deforestation, or afforestation) on large-scale AGB change.
**5 Conclusions**
DGVMs are useful tools for simulation of regional- and global-scale carbon



dynamics. In this research, we evaluated the model performance in modeling global
carbon dynamics after calibration of IBIS using in situ GPP and plot-level AGB data
collected from the literature. Independent validation showed that IBIS can reproduce
GPP and ET with acceptable accuracies at the site and global levels. On the
global-scale, IBIS simulation of total AGB gave results similar to those obtained in
other studies. However, discrepancies were observed between model-derived and
observed spatial patterns of AGB for Amazonian forests, mainly because of the
unique parameter set used in the model. Two metrological datasets, i.e., Princeton and
CRU, were used to test the model uncertainties caused by climate drivers. The results
indicated that the two meteorological inputs give substantially different global-scale
AGBs. Further analysis showed that this difference was small compared with the
parameter-induced difference. The conclusions of our research highlight the necessity
of considering the heterogeneity of key model physiological parameters in modeling
global AGB. The research also shows that to simulate large-scale carbon dynamics,
both carbon flux and AGB data are necessary to constrain the model. The main
conclusions of our research could help to improve model simulation of the global
carbon cycle.

**Acknowledgements**
This study was financially supported by the National Science Foundation of
China (Grant No. 41301020) and the National Key Basic Research Program of China
(2013CB956604). We are grateful to the PIs and Co-Is of FLUXNET who make their
data freely available to the ecological modelling community through the FLUXNET





archive (http://fluxnet.ornl.gov/), in particular by the following networks: AmeriFlux
(U.S. Department of Energy, Biological and Environmental Research, Terrestrial
Carbon Program (DE-FG02-04ER63917 and DE-FG02-04ER63911)), AfriFlux,
AsiaFlux, CarboAfrica, CarboEuropeIP, CarboItaly, CarboMont, ChinaFlux,
Fluxnet-Canada (supported  by CFCAS, NSERC, BIOCAP, Environment  Canada,
and NRCan), GreenGrass, KoFlux, LBA, NECC, OzFlux, TCOS-Siberia, USCCC.
We acknowledge the financial support to the eddy covariance data harmonization
provided  by CarboEuropeIP, FAO-GTOS-TCO, iLEAPS, Max  Planck  Institute  for
Biogeochemistry, National  Science  Foundation, University  of  Tuscia, Université
Laval, Environment  Canada and US  Department  of  Energy and  the  database
development and technical support from Berkeley Water Center, Lawrence Berkeley
National Laboratory, Microsoft Research eScience, Oak Ridge National Laboratory,
University of California – Berkeley and the University of Virginia.





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





## Tables

**Table 1** Key PFT-dependent parameters for IBIS calibration. The abbreviations are defined as follows: vmax_pft: maximum Rubisco capacity at top of canopy (µmol m$^{-2}$ s$^{-1}$); SLA: specific leaf area (m$^2$ kg$^{-1}$); $\tau_l$: residence time of foliar biomass (years); $\tau_r$: residence time of root biomass (years); $\tau_w$: residence time of wood biomass (years); $a_{leaf}$: allocation coefficient of total photosynthate in foliar biomass (fraction); $a_{root}$: allocation coefficient of total photosynthate in root biomass (fraction); $a_{wood}$: allocation coefficient of total photosynthate in wood biomass (fraction); $P_{min}$: monthly minimum precipitation (mm month$^{-1}$); $T_{minL}$: absolute minimum temperature (lower limit, ℃); $T_{minU}$: absolute minimum temperature (upper limit, ℃); $T_{warm}$: temperature of the warmest month ( ℃) (C$_4$ plants only); GDD: minimum growing degree days above 5 ℃ threshold for upper-canopy types; minimum growing degree days above 0 ℃ threshold for lower-canopy types. The plant functional type (PFT) numbers defined in IBIS are as follows: 1, tropical broadleaf evergreen trees; 2, tropical broadleaf drought-deciduous trees; 3, warm–temperate broadleaf evergreen trees; 4, temperate conifer evergreen trees; 5, temperate broadleaf cold-deciduous trees; 6, boreal conifer evergreen trees; 7, boreal broadleaf cold-deciduous trees; 8, boreal conifer cold-deciduous trees; 9, evergreen shrubs; 10: cold-deciduous shrubs; 11, warm (C4) grasses; and 12, cool (C3) grasses.

| PFT | vmax_pft | SLA | $\tau_l$ | $\tau_r$ | $\tau_w$ | $a_{leaf}$ | $a_{root}$ | $a_{wood}$ | $P_{min}$ | $T_{minL}$ | $T_{minU}$ | $T_{warm}$ | GDD |
|---|---|---|---|---|---|---|---|---|---|---|---|---|---|
| 1 | 55 | 25 | 1.01 | 1 | 60 | 0.3 | 0.3 | 0.4 | >5.0 | >0.0 | – | – | – |
| 2 | 45 | 25 | 1 | 1 | 60 | 0.3 | 0.3 | 0.4 | – | >0.0 | – | – | – |
| 3 | 40 | 25 | 1 | 1 | 25 | 0.3 | 0.3 | 0.4 | – | >−5.0 | <0.0 | – | – |
| 4 | 30 | 12.5 | 2 | 1 | 35 | 0.3 | 0.4 | 0.3 | – | >−45.0 | <0.0 | – | >1100 |
| 5 | 40 | 25 | 1 | 1 | 35 | 0.3 | 0.3 | 0.4 | – | >−45.0 | <0.0 | – | >1100 |
| 6 | 25 | 12.5 | 2.5 | 1 | 52 | 0.3 | 0.4 | 0.3 | – | >−57.5 | <−45.0 | – | >350 |
| 7 | 30 | 25 | 1 | 1 | 52 | 0.3 | 0.3 | 0.4 | – | >−57.5 | <−45.0 | – | >350 |
| 8 | 35 | 25 | 1 | 1 | 52 | 0.3 | 0.3 | 0.4 | – | – | <−45.0 | – | >350 |
| 9 | 27.5 | 12.5 | 1.5 | 1 | 5 | 0.45 | 0.4 | 0.15 | – | – | – | – | >100 |
| 10 | 27.5 | 25 | 1 | 1 | 5 | 0.45 | 0.35 | 0.2 | – | – | – | – | >100 |
| 11 | 15 | 20 | 1.25 | 1 | – | 0.45 | 0.55 | 0 | – | – | – | >22.0 | >100 |
| 12 | 25 | 20 | 1.5 | 1 | – | 0.45 | 0.55 | 0 | – | – | – | – | >100 |



**Table 2** Comparison of observed and model-derived gross primary production (GPP; gC m$^{-2}$ month$^{-1}$)
and evapotranspiration (ET; mm month$^{-1}$) for 39 sites. The regression coefficients of slope (*a*),
intercept (*b*), $R^2$, and root-mean-square error (RMSE) deviations are also shown. The PFT definitions
are the same as in Table 1.

| Longitude | Latitude | Site | PFT | GPP (gC m$^{-2}$ month$^{-1}$) | | | | ET (mm m$^{-2}$ month$^{-1}$) | | | |
|---|---|---|---|---|---|---|---|---|---|---|---|
| | | | | *a* | *b* | $R^2$ | RMSE | *a* | *b* | $R^2$ | RMSE |
| 131.15 | -12.49 | Au-How | 2 | 1.54 | -64.90 | 0.73 | 50.13 | 1.06 | -37.31 | 0.54 | 39.18 |
| -68.75 | 45.21 | US-Ho2 | 4 | 1.11 | -2.42 | 0.97 | 18.95 | 0.99 | 8.40 | 0.93 | 8.61 |
| -121.56 | 44.45 | US-Me2 | 4 | 0.74 | 6.91 | 0.93 | 16.44 | 0.56 | 5.56 | 0.67 | 10.68 |
| -121.61 | 44.32 | US-Me3 | 4 | 1.14 | 14.96 | 0.91 | 18.20 | 0.72 | 9.19 | 0.71 | 10.22 |
| -121.57 | 44.44 | US-Me5 | 4 | 1.18 | 12.55 | 0.90 | 18.43 | 0.72 | 7.10 | 0.73 | 9.13 |
| -76.67 | 35.80 | US-NC2 | 4 | 0.64 | 62.28 | 0.85 | 25.13 | 0.78 | -1.89 | 0.92 | 9.81 |
| -105.55 | 40.03 | US-NR1 | 4 | 0.53 | 11.09 | 0.70 | 22.29 | 0.52 | 4.65 | 0.59 | 11.75 |
| -89.87 | 34.25 | US-Goo | 5 | 0.54 | 128.53 | 0.48 | 53.27 | 0.69 | 19.37 | 0.65 | 18.71 |
| -72.17 | 42.54 | US-Ha1 | 5 | 0.65 | 67.39 | 0.69 | 59.59 | 0.86 | 12.60 | 0.86 | 11.72 |
| -72.19 | 42.54 | US-LPH | 5 | 0.56 | 82.25 | 0.57 | 73.94 | 0.67 | 20.40 | 0.78 | 15.00 |
| -86.41 | 39.32 | US-MMS | 5 | 0.67 | 89.27 | 0.70 | 53.94 | 0.73 | 17.00 | 0.87 | 12.16 |
| -92.20 | 38.74 | US-MOz | 5 | 0.60 | 86.36 | 0.69 | 50.23 | 0.60 | 19.84 | 0.69 | 19.75 |
| -82.24 | 29.76 | US-SP2 | 5 | 0.27 | 160.04 | 0.25 | 36.50 | 0.74 | 21.01 | 0.61 | 20.95 |
| -84.29 | 35.96 | US-WBW | 5 | 0.52 | 113.58 | 0.61 | 51.65 | 0.81 | 18.54 | 0.90 | 10.70 |
| -98.48 | 55.88 | CA-NS1 | 6 | 1.80 | 13.11 | 0.87 | 36.49 | 1.04 | 0.21 | 0.77 | 13.33 |
| -98.52 | 55.91 | CA-NS2 | 6 | 1.41 | 24.90 | 0.56 | 65.64 | 1.22 | -1.71 | 0.85 | 11.07 |
| -98.38 | 55.91 | CA-NS3 | 6 | 1.65 | 17.76 | 0.77 | 47.40 | 1.10 | -0.11 | 0.87 | 10.33 |
| -98.38 | 55.91 | CA-NS4 | 6 | 2.59 | 12.25 | 0.95 | 21.31 | 1.80 | 0.80 | 0.89 | 8.93 |
| -98.49 | 55.86 | CA-NS5 | 6 | 1.53 | 12.36 | 0.94 | 23.15 | 1.11 | 0.56 | 0.94 | 6.88 |
| -99.95 | 56.64 | CA-NS7 | 6 | 2.31 | 45.05 | 0.72 | 56.61 | 1.04 | 0.56 | 0.83 | 11.21 |
| -121.95 | 45.82 | US-Wrc | 6 | 0.93 | -7.75 | 0.81 | 34.30 | 0.82 | 2.63 | 0.64 | 14.71 |
| -89.98 | 46.08 | US-Los | 7 | 0.74 | 98.07 | 0.52 | 75.02 | 0.93 | 9.90 | 0.89 | 11.06 |
| -89.35 | 46.24 | US-Syv | 7 | 0.91 | 38.40 | 0.83 | 46.36 | 1.04 | 8.47 | 0.95 | 7.63 |
| -90.08 | 45.81 | US-WCr | 7 | 0.78 | 54.64 | 0.75 | 59.25 | 0.90 | 12.99 | 0.90 | 10.85 |
| -110.51 | 31.59 | US-Aud | 10 | 0.82 | 49.34 | 0.49 | 42.11 | 1.23 | -2.36 | 0.70 | 18.77 |
| -155.75 | 68.49 | US-Ivo | 10 | 1.65 | 9.08 | 0.57 | 27.83 | 0.80 | 7.20 | 0.62 | 10.14 |
| -80.67 | 28.61 | US-KS2 | 10 | 0.45 | 148.76 | 0.36 | 25.63 | 0.91 | 29.25 | 0.67 | 18.65 |
| -116.64 | 33.38 | US-SO4 | 10 | 1.32 | 37.85 | 0.31 | 42.54 | 0.35 | 18.28 | 0.07 | 20.30 |
| -120.95 | 38.41 | US-Var | 10 | 0.62 | 59.95 | 0.72 | 32.77 | 0.65 | 16.55 | 0.75 | 10.18 |
| -98.04 | 35.55 | US-ARb | 12 | 0.27 | 103.38 | 0.34 | 45.75 | 0.74 | 18.17 | 0.76 | 18.63 |
| -98.04 | 35.55 | US-ARc | 12 | 0.32 | 98.57 | 0.32 | 46.30 | 0.68 | 17.94 | 0.71 | 20.22 |
| -97.49 | 36.61 | US-ARM | 12 | 0.80 | 106.17 | 0.28 | 57.43 | 1.28 | 14.59 | 0.51 | 28.55 |
| -96.84 | 44.35 | US-Bkg | 12 | 1.05 | 62.24 | 0.59 | 69.07 | 0.69 | -2.61 | 0.77 | 19.17 |
| -88.29 | 40.01 | US-Bo1 | 12 | 0.33 | 123.55 | 0.26 | 88.55 | 0.81 | 6.18 | 0.82 | 14.56 |
| -105.10 | 48.31 | US-FPe | 12 | 0.68 | 44.69 | 0.18 | 56.25 | 0.53 | 13.15 | 0.48 | 19.89 |
| -93.09 | 44.71 | US-Ro1 | 12 | 0.37 | 116.91 | 0.25 | 102.30 | 0.90 | 5.45 | 0.88 | 13.48 |
| -93.09 | 44.72 | US-Ro3 | 12 | 0.50 | 96.89 | 0.40 | 91.07 | 0.91 | 8.14 | 0.81 | 16.91 |
| -109.94 | 31.74 | US-Wkg | 12 | 1.22 | 44.34 | 0.25 | 67.95 | 1.51 | -3.65 | 0.51 | 26.96 |
| -96.86 | 37.52 | US-Wlr | 12 | 0.78 | 86.50 | 0.65 | 47.84 | 0.99 | 6.83 | 0.72 | 22.56 |

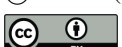



**Table 3** Comparison of model-derived forest carbon density (Mg C ha$^{-1}$) with those from other studies.
Pan et al. (2001) calculated carbon densities for both above- and below-ground biomass. Numbers in
brackets for Pan et al. (2011) show the AGB values assuming that the AGB accounts for 80% of total
biomass density.

| Source | Method | Forest Carbon Density (Mg C ha$^{-1}$) | | | Carbon Stock (Pg) |
|---|---|---|---|---|---|
| | | Europe | North America | Global | Global |
| Goodale et al.(2002) | Forest Inventory | 38.8 | 44.6 | | |
| Liski et al.(2003) | Forest Inventory | 43 | 43 | | |
| Thurner et al.(2014) | Remote Sensing | 60.8 ±22.4 | 45.3 ±17.1 | | |
| Pan et al.(2011) | Forest Inventory | 60.5 (48.4) | 68.7 (54.9) | 94.2 (75.4) | 362.6 (290.1) |
| This Study | Model | 59.24 ±20.04 | 53.74 ±36.39 | 82.96 | 276.5 |



**Figure captions**
Fig. 1 Taylor diagram of (a) GPP (gC m$^{-2}$ year$^{-1}$) and (b) ET (mm year$^{-1}$) for 39 flux
towers.

Fig. 2 Comparison of annual observed and modeled values for (a) GPP (gC m$^{-2}$ year$^{-1}$)
and (b) ET (mm year$^{-1}$), and (c) independent validation of GPP on annual scale. The
dashed line shows the 1:1 line.

Fig. 3 (a) Modeled GPP (gC m$^{-2}$ year$^{-1}$) averaged for 2000–2011 and (b) difference
between modeled value and that reported by Jung et al. (2011).

Fig. 4 (a) Modeled ET (mm year$^{-1}$) averaged for 2000–2011 and (b) difference between
modeled value and that reported by Jung et al. (2011).

Fig. 5 Comparison of annual observed and modeled values for (a) site-year AGB (Mg
ha$^{-1}$) and (b) different sites, and (c) independent validation. The dashed line shows the
1:1 line.

Fig. 6 (a) Modeled global patterns of AGB (Mg ha$^{-1}$) averaged for 2000–2010 and (b)
latitudinal AGB patterns.



Fig. 7 (a) Comparison of observed AGB (left panel, points show plot locations) and
difference between modeled and observed AGBs for Amazonian forests (right panel), and
(b) relative error [(Modeled − observed)/observed $\times 100\%$] frequency.

Fig. 8 (a) Difference between model-derived AGB driven by Princeton and CRU
meteorological datasets. The Princeton and CRU data are on daily and monthly
timescales, respectively.





**Figure 1**

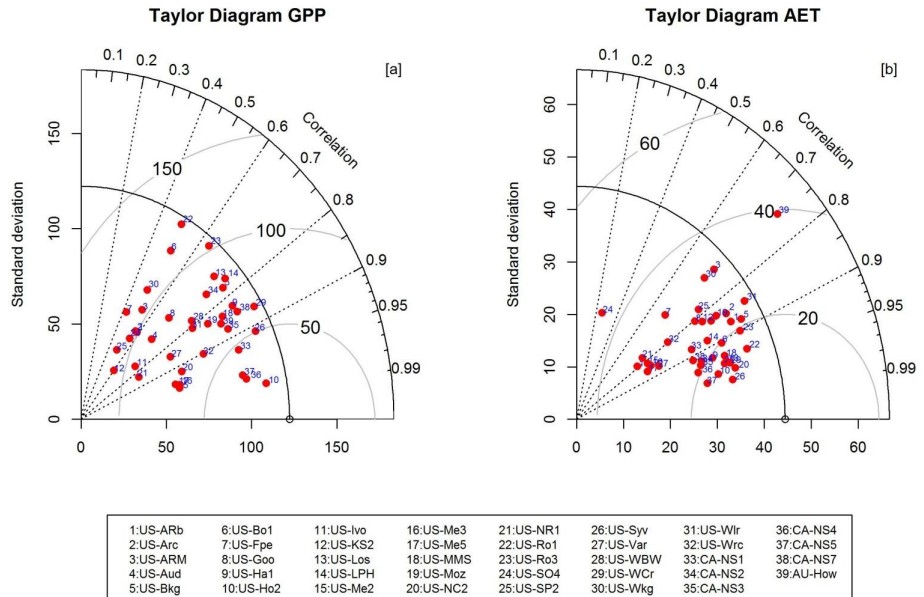






**Figure 2**

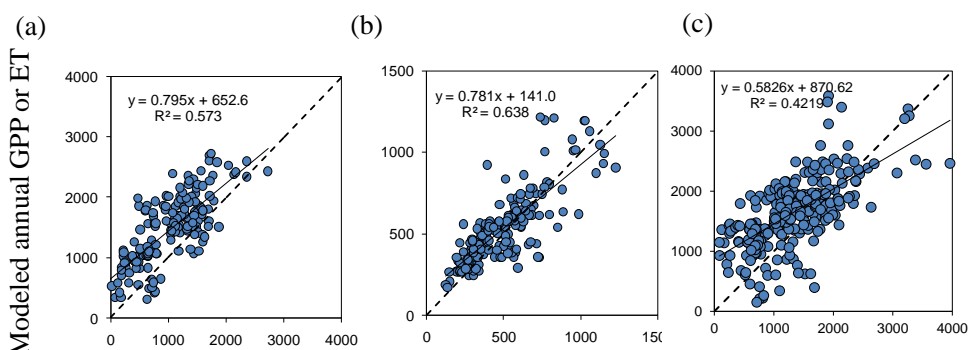






**Figure 3**

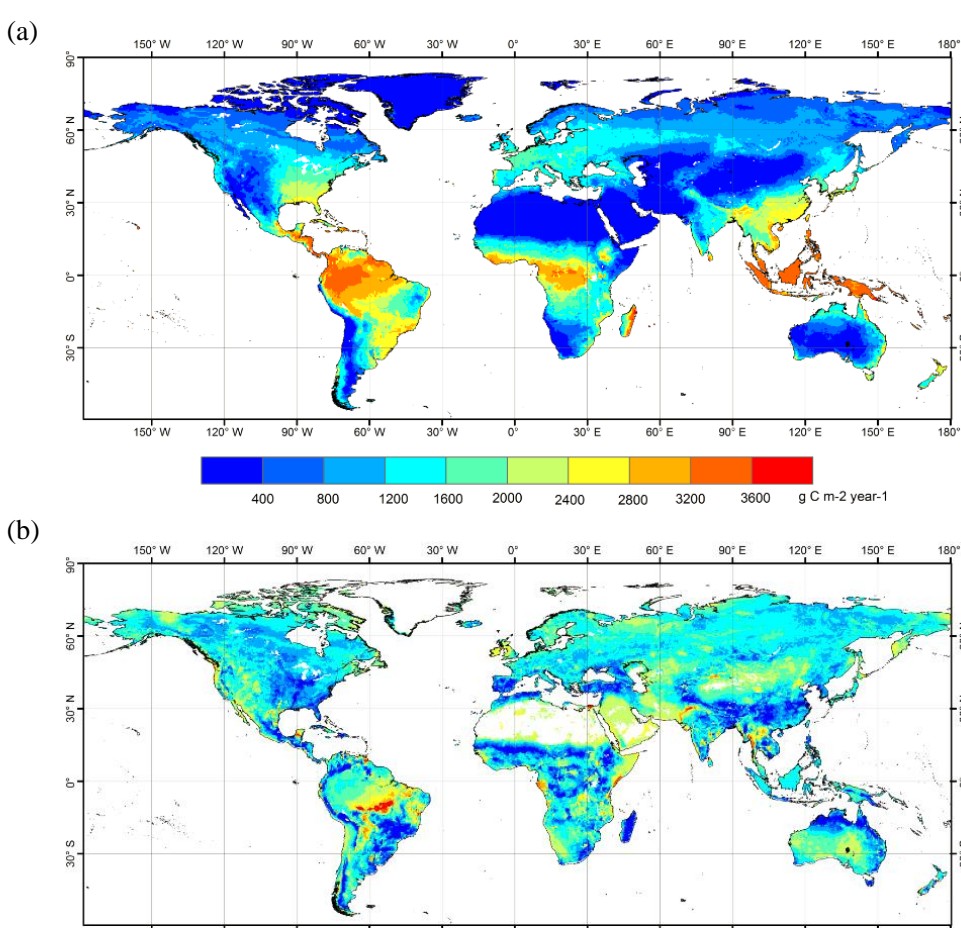






**Figure 4**

(a)

(b)

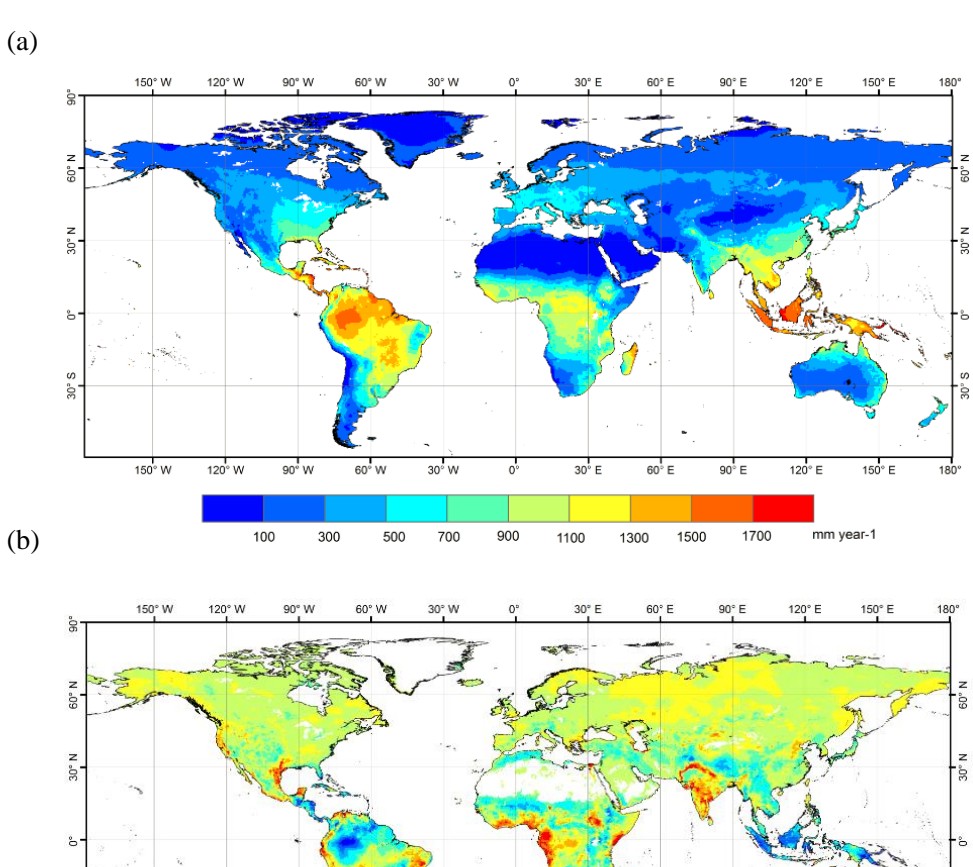






**Figure 5**

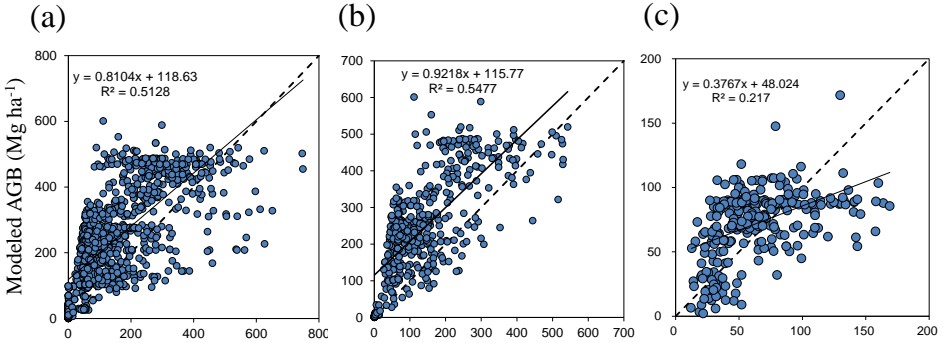

Observed above ground biomass (AGB, Mg ha⁻¹)






**Figure 6**

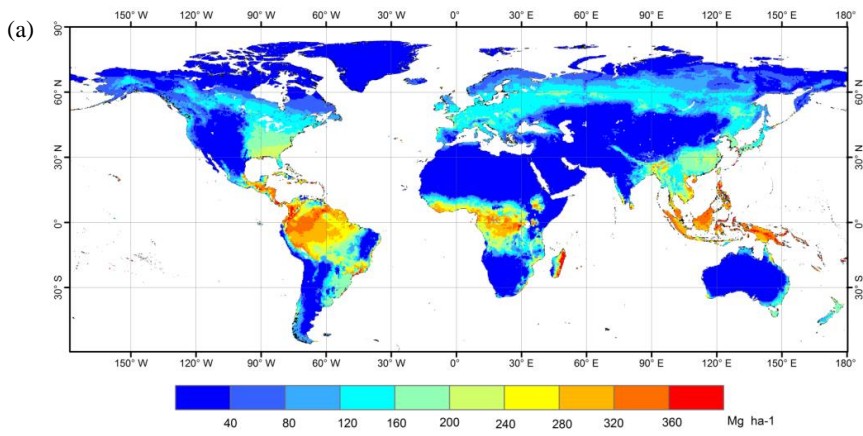

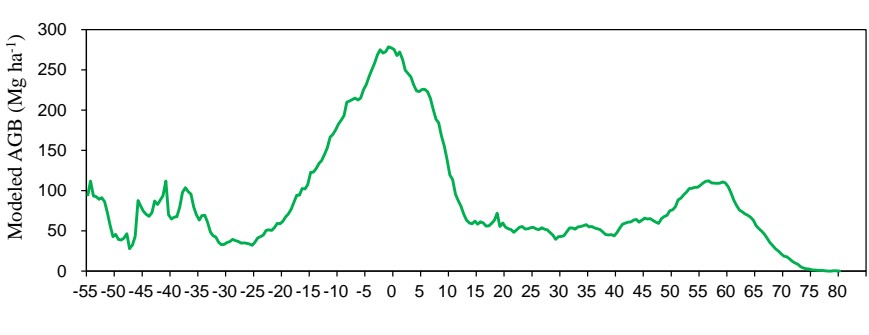






**Figure 7**

(a)

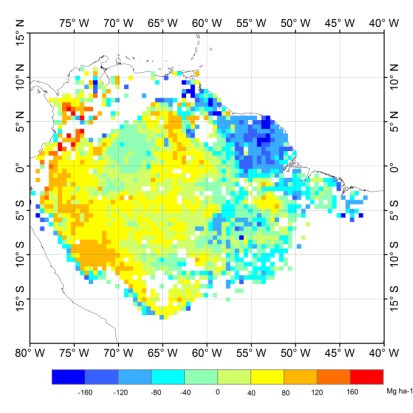

(b)




**Figure 8**

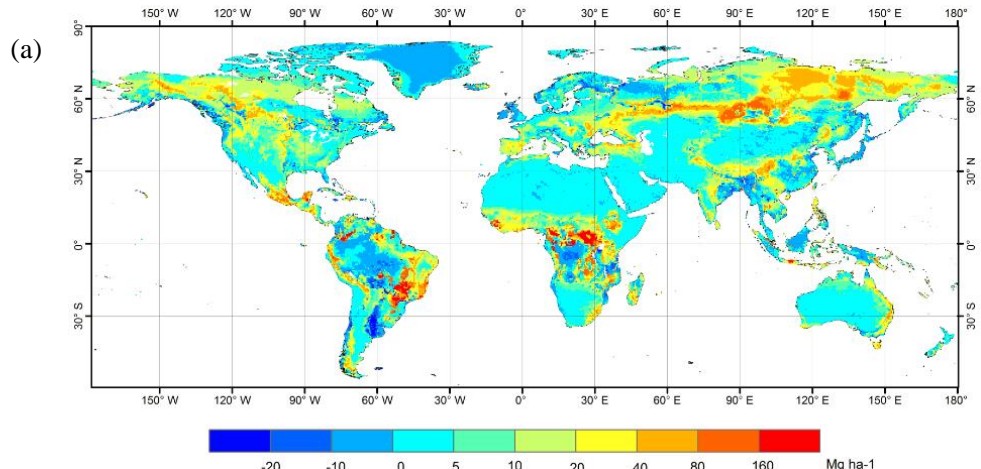

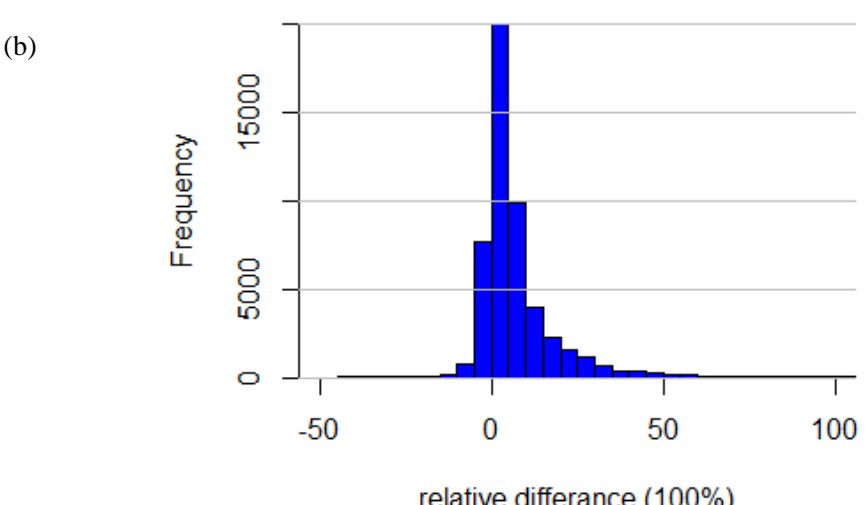
