# Peer review of "Published: 25 April 2016"

_Biogeosciences, 2016_

## Referee Comment (RC1) · Anonymous Referee #1 · 29 Apr 2016

This study uses IBIS in a calibration then validation study to examine AGB, GPP, and ET. This is not a very novel contribution but has the potential to be a solid iteration. My main concerns are (1) the lack of detail on calibration, (2) the lack of a pre-calibration baseline, (3) the lack of variable tau even though this is known and has already been done for this model, (4) the relevance of ET, (5) the lack of detail on the flux data, and (6) language issues. These are all detailed in the specific comments below. Assuming these are addressed this study could be suitable for publication in Biogeosciences.

L24: You go from "the simulation of carbon dynamics on regional and global scales" to biomass. But carbon dynamics are more than biomass! Also, later in the paper you look at ET, this is not mentioned in the abstract until you get to "Independent validation"

[Figure]

(L31).

L40: I'm having a hard time with your: "The conclusions of our research highlight the necessity of considering the heterogeneity of key model physiological parameters in modeling global AGB." This is not a new insight. What have you added here? How has the body of knowledge on this point been expanded? And I think the community knows that getting fluxes and stocks correct is a good idea to enhance predictive skill. More generally, I am struggling with the novelty of this study. I feel like I've read dozens of such papers before. . .

L54: One of many language issues: "The large carbon stock in the terrestrial ecosystem indicates the need for a reliable description of its current distribution and prediction of future variations". This is a non sequitur as written. I think I know what you want to say here but I should not have to divine intent and meaning. But please have this edited by a native speaker.

L57: Is not the main reason it is hard to get global scale carbon stocks a lack of good observations?

L73: Try "alter global biomass". The full sentence here (L72) is also a non sequitur as written.

L87: Here you only mention potential whereas above you also had present-day. Please be careful! Trivially we are nowhere near some idealized potential state such that a model might be more concerned with getting present-day correct.

L100: Starting here in this paragraph I feel you have already made these points. Also, where is ET. It's thrown in at seemingly random times and is not part of the story. Why do you even include ET?

L116: I need more detail on how you calibrated IBIS parameters. It's not explained to sufficient detail anywhere in the text.

L122: You mean hydrological?

L125: Remove "In detail,".

L143: Remove both uses of "the".

L191: Is not there a more recent version of CRUTS?

L197: "The Princeton..." reads awkwardly. Maybe "The Princeton forcing data..." is better.

L207: If IBIS can't do croplands why perform any comparison?

L214: Same awkward "In detail, . . .". Perhaps you want "In brief, ..." instead.

L216: A minor point but you might want to capitalize Random Forest as you use forest as a PFT/biome in many places.

L219: I am still unsure if you are doing global runs here? The preceding text references the Amazon. Also, despite your description I am unclear how "plot-level AGB data from the literature" was used to tweak IBIS out-of-the-box parameters. That is, how was Objective (2) satisfied?

L221: Break into 2 sentences please.

L228: no "the" before GPP.

L232: a rho of 0.6 means 36% variance explained. And this is a lower bound? Can you elaborate here? How can you reconcile the lower bound you use in the text with your "the model can simulate the energy balance well" (L236) statement.

L245: How does a small value have a large intercept?

L247: Try "When GPP was below..."

L248: Try "simulated GPP was. . ." GPP is not used as a plural. Fix this throughout the paper.

L251: How does a tower focus on high-production ecosystems when it's sited in a

marginal system? This makes no sense.

L252: I found this section rather hard to read. One example: "the model simulation results are around 100 mm year−1 smaller than those from Jung et al. (2011), especially for low ET areas (Fig. 4)." The "especially" fragment makes no sense. And why "model simulation results" as opposed to just "simulated results"? I would encourage you to look at the full manuscript for clarity and flow.

L254: Remove the comma.

L277: Your "scale difference" argument here is not clear. Please elaborate.

L279: What is "plot accuracy"?

L286: In this section you overuse the word "shows".

L292: What is "below −30°S"? The minus sign is redundant here.

L294: How did you pick the bounds for your latitudinal regressions? And why is this relevant here? In general, the writing has a tendency to walk the reader through the figures. That's not bad per se but it's overly verbose as is. Let the figures do some of the talking. I would recommend less length here.

L312: Language again, the hanging fragments of "according to the two meteorological datasets" and "with large heterogeneities in different areas" simply do not add value, only volume.

L325: "Comparisons of carbon densities have the advantage over AGB comparisons that they eliminate the uncertainties induced by global vegetation areas used in different studies." Not sure I buy this. It seems you are assuming that density is the same across vegetation gradients. And we know this is not the case. And your L336 goes against this anyway.

L361: There has been more work on tau beyond the IBIS work. And if this is known as the "most important parameter in determining the spatial variance" why was it not

addressed here?

L374: "The interannual variations in the GPP were mainly caused by different meteorological drivers." This is an odd sentence. How is IAV "caused" by different forcings? And drop "the" before GPP.

L379: Figure 8 does not show this. There is no "transfer function" from GPP to AGB (or vice versa) shown in the figure!

L422: "The research also shows that to simulate large-scale carbon dynamics, both carbon flux and AGB data are necessary to constrain the model." Fine, I think we knew this before this study was executed. But why not extend this. Add results using the out-of-the-box parameter values, so pre-calibration. What is that effect size? How does parameter-induced spread compare with forcing data-induced spread and spatial heterogeneity. Your second question (L116) is: "Can a single set of calibrated parameters accurately map the patterns of GPP and AGB?" It strikes me that to answer this query we need a baseline without calibration.

Table 2: Please add time period so we have a better sense of sample size. How did you navigate the La Thuile fair use data policy? I'm assuming that's where you harvested the FLUXNET data.

Figure 1: Label x-axis. What are the 20, 40, 60, 50, 100, 150 values for?

Figure 2: Add labels for y- and x-axis. Get rid of the "or". Make sure inset text does not overplot.

Figure 3: Minor point but the use of the same colormap in both plots (one inverted, one not) makes this harder to read than needed. A third panel showing relative differences(100*b/a in effect) would be a good add. Fix subscripts on colorbar units.

Figure 4: Same comments as for Figure 3.

Figure 5: Same comments as for Figure 2.

Figure 6: Fix subscript. I'd like to see (b) flipped so that the lat bands match how one would read map. Why not add a mean zonal vector for both forcing cases?

Figure 7: The (c) panel occupies a bit too much space. It's really not that exciting anyway, you could drop it. Also, ylabel is misspelt. Or, map the relative differences as I've suggested in preceding figures.

Figure 8: Same comment as for Figure 7.

---

## Referee Comment (RC2) · Anonymous Referee #2 · 23 May 2016

Though I would like to be encouraging of work in the general direction of confronting ecosystem process models with emerging data, unfortunately this effort does not provide a good example. There are many problems with this work including many detailed below, but the biggest problem is that the findings, interpretations, and conclusions are not at all supported by the work that has been done (see V. below).

I. Model Calibration Is Not Described: The paper suggests that it performs a model calibration but there is no information on this. A set of model (IBIS) parameters are apparently calibrated with flux tower data on GPP and ET from select sites and with plot-level aboveground biomass data. However, there is no description of the model calibration, and no parameter uncertainty or parameter correlation (equifinality) analy-

[Figure]

sis.

II. Data Sources Are Not Disclosed: The paper does not cite its data source(s) for the plot-level aboveground biomass dataset that it apparently used for calibration (though maybe just for evaluation?). It is suggested that most of the data come from China, though the Figure S1 shows a broad global distribution. Individual citations for all data sources must be provided, and the methods must explain the methods of data collection for each of those sources. It's inadequate to simply say "from the literature" and show a map of locations.

III. Model SetUp Incompletely Described: There is no description of the modeling procedure. Was a spin-up performed to bring carbon pools to some equilibrium state? How were PFTs assigned to grid cells, and/or does the model simulate PFT distributions that match with the other datasets? There is a risk of the wrong PFTs being simulated, for example where land use has substantially altered the PFT from a model-estimated dominant PFT (e.g. if deforestation removed trees with grasses, crops, or savanna instead).

IV. Model Evaluation (Simply Comparing to Data) Does Not Go Far Enough: This paper's main point is that new datasets need to be used to confront models and improve them. However, the paper offers nothing to improve the model that is used. Discrepancies are shown but there is no new insight about why, or how the model structure or parameters would best be modified to come to resolution with the data, where appropriate.

V. Findings and Conclusions Do Not Follow from Results and Do Not Advance Science in a Useful Way: The paper purports to show the following but each is poorly substantiated if at all.

1) Claim: Results of a DGVM can be sensitive to the meteorological driver data that are used but that parameter uncertainties are more important. Concern: This is known, and in fact is not precisely shown here. The paper does not compare sensitivity to

parameter values in any way and thus cannot make this claim.

2) Claim: Bias or error in GPP caused by meterological data can be transferred to AGB carbon stock. Concern: This is already known, and in fact is not precisely shown here (paper does not show that GPP bias or error relates to AGB bias or error).

3) Claim: To improve model accuracy, modelers should pay attention to both model parameter calibration and meterological drivers, with a focus on the former. Concern: This is known, and again, is not evidenced by anything in the present study.

4) Claim: DGVMs are useful tools for simulation of regional- and global-scale carbon dynamics. Concern: No doubt they are but this is not a conclusion of the study.

5) Claim: Discrepances were observed between model-derived and observed spatial patterns of AGB for Amazonian forests, mainly because of the unique parameter set used in the model. Concern: Only a single parameter set was tested so you cannot claim that that is the source of the mismatch. Model structure could be a source of mismatch. Meteorological driver data could too. Nothing presented supports this claim.

6) Claim: The conclusions of our research highlight the necessity of considering heterogeneity of key model physiological parameters in modeling global AGB. Concern: This is already well established and not at all demonstrated by the present study.

7) Claim: The research also shows that to simulate large-scale carbon dynamics, both carbon flux and AGB data are necessary to constrain the model. Concern: There is nothing here to support this claim. The study does constrain with C flux only, with AGB data only, and then with both to show that both are needed to recover key metrics of carbon dynamics. This claim is another throw away with no substance in the current paper.

VI. It is Unclear Why FLUXNET Upscaled Product Is Newly Estimated: The calibrated model is then compared against a flux-tower upscaled GPP and ET product (Jung et al. 2011) but was actually re-estimated here for some unknown reason, and came up

with substantially different results.

VII. Study Involves a New Phenology Model That is Untested with No Evaluation: When you introduce a new model component such as the phenology model used here it is fitting to evaluate if that model component performs well compared to data. This is, in fact, part of the point of the paper, however the idea seems to have been missed with respect to this paper's new implementation of the phenology component in IBIS.

VIII. This paper does not appear to adhere to the FLUXNET Data Fair Use Policy. It does not cite the appropriate papers and does not include appropriate acknowledgement.

IX. References missing: e.g. Stockli et al. 2008 is not in the references list, maybe others.

---

## Author Comment (AC1) · 19 Jun 2016

Thank you for the comments on our manuscript. The response letter are provided in PDF files. We also upload the revised manuscript (including figures) as Supplement.

Please also note the supplement to this comment: http://www.biogeosciences-discuss.net/bg-2016-142/bg-2016-142-AC1-supplement.zip

---

## Author Comment (AC2) · 19 Jun 2016

Thank you for your comments. We now provided the response letter and revised manuscript(including figures) as Supplement.

Please also note the supplement to this comment:
http://www.biogeosciences-discuss.net/bg-2016-142/bg-2016-142-AC2-supplement.zip
* * *